

# Evaluation of female *Aedes aegypti* proteome via LC-ESI-MS/MS using two protein extraction methods

Abubakar Shettima[1,2], Intan H. Ishak[3,4], Syahirah Hanisah Abdul Rais[1], Hadura Abu Hasan[3,4] and Nurulhasanah Othman[1]

[1] Institute for Research in Molecular Medicine (INFORMM), Universiti Sains Malaysia, Gelugor, Pulau Pinang, Malaysia
[2] Department of Microbiology, Faculty of Science, University of Maiduguri, Maiduguri, Borno State, Nigeria
[3] School of Biological Sciences, Universiti Sains Malaysia, Gelugor, Pulau Pinang, Malaysia
[4] Vector Control Research Unit (VCRU), School of Biological Sciences, Universiti Sains Malaysia, Gelugor, Pulau Pinang, Malaysia

## ABSTRACT

**Background.** Proteomic analyses have broadened the horizons of vector control measures by identifying proteins associated with different biological and physiological processes and give further insight into the mosquitoes' biology, mechanism of insecticide resistance and pathogens-mosquitoes interaction. Female *Ae. aegypti* ingests human blood to acquire the requisite nutrients to make eggs. During blood ingestion, female mosquitoes transmit different pathogens. Therefore, this study aimed to determine the best protein extraction method for mass spectrometry analysis which will allow a better proteome profiling for female mosquitoes.

**Methods.** In this present study, two protein extractions methods were performed to analyze female *Ae. aegyti* proteome, via TCA acetone precipitation extraction method and a commercial protein extraction reagent CytoBuster™. Then, protein identification was performed by LC-ESI-MS/MS and followed by functional protein annotation analysis.

**Results.** The CytoBuster™ reagent gave the highest protein yield with a mean of 475.90 μg compared to TCA acetone precipitation extraction showed 283.15 μg mean of protein. LC-ESI-MS/MS identified 1,290 and 890 proteins from the CytoBuster™ reagent and TCA acetone precipitation, respectively. When comparing the protein class categories in both methods, there were three additional categories for proteins identified using CytoBuster™ reagent. The proteins were related to scaffold/adaptor protein (PC00226), protein binding activity modulator (PC00095) and intercellular signal molecule (PC00207). In conclusion, the CytoBuster™ protein extraction reagent showed a better performance for the extraction of proteins in term of the protein yield, proteome coverage and extraction speed.

Corresponding author
Nurulhasanah Othman, nurulhasanah@usm.my

## INTRODUCTION

Proteomics is a fast-developing research field. Scientists have experimented continuously to profile and catalogue insect proteome, including mosquitoes at different tissues and organelle and in varying physiological states (*Shashank & Haritha, 2014*). Furthermore, their interactions with viruses, parasites and toxins were also studied (*Shashank & Haritha, 2014*).

In general, mosquito proteomic analyses have revealed features of haemolymph proteins, midgut peritrophic matrix proteins, mosquito-head proteins during different feeding (sugar or blood) (*Shashank & Haritha, 2014*; *Whiten et al., 2018*). Proteomics using LC-MS/MS (Liquid Chromatography-Mass Spectrometry) also made possible the elucidation of host-virus interactions (*Shashank & Haritha, 2014*). Female *Ae. aegypti* ingest human blood to acquire the nutrients necessary to produce eggs. During blood ingestion, female mosquitoes may transmit different pathogens, including viruses such as dengue and yellow fever to their respective host.

Several studies have investigated differential protein expressions in mosquitoes to elucidate protein regulations during various physiological conditions (*Popova-Butler & Dean, 2009*; *Djegbe et al., 2011*; *Cancino-Rodezno et al., 2012*; *Wang et al., 2015*; *Whiten et al., 2018*; *Mano et al., 2019*). Protein functional classification analysis also reveals their biological processes, molecular function, and cellular components as well as phylogenic and ancestral strings by the functional annotation and detailed bioinformatics analysis (*Shashank & Haritha, 2014*).

It is critical to determine which extraction protocol produce high protein yield and achieve more protein hits from protein identification by LC-ESI-MS/MS analysis. Therefore, this study compared two extraction methods for protein identification analysis in female *Ae. aegypti*, by following a TCA acetone precipitation method by *Wang et al. (2015)* and a commercial protein extraction reagent CytoBuster™ (Sigma, Germany) to assess the performance of the extraction method for LC-ESI-MS/MS analysis which will allow a better proteome profiling for female mosquitoes.

## MATERIALS & METHODS

### Mosquito samples

*Ae. aegypti* eggs papers were obtained from the Vector Control Research Unit (VCRU), Universiti Sains Malaysia (USM), and reared at the Insectary of the School of Biological Sciences, Universiti Sains Malaysia, at a constant room temperature of approximately 28 °C and 75% relative humidity. Eggs were submerged in water and applied a larvae food composed of grounded dog biscuit, beef liver, powdered milk and yeast in a ratio of 2:1:1:1. Pupae from the larvae were transferred into half-full disposable cups and placed in a cage net for adult mosquitoes to emerge. Adult mosquitoes were fed with 10% sucrose solution before harvesting and 3–5 days old female mosquitoes were harvested for the protein extraction at the Institute for Research in Molecular Medicine (INFORMM), USM.

## Protein extraction

Three biological replicates comprised of 20 adult female mosquitoes in each replicate, were analyzed for each extraction method. For the TCA acetone precipitation method (*Wang et al., 2015*), female adult *Ae. aegypti* were washed three times with distilled deionized water to remove food particles and molted skin before homogenization at 50 rpm for 5–10 mins in a mini bead beater using 0.5 mm zirconia beads in 10% TCA cold acetone and 10 mM DTT, and incubated overnight at $-20$ °C. Then centrifuged at 4 °C, $15,000 \times g$ for 5 mins and the pellets resuspended in lysis buffer (7M urea, 2M Thiourea, 4% CHAPS) containing 1 mM PMSF, 2 mM EDTA and 10 mM DTT. The mixture was sonicated for 1 min, with 10-sec pulse and 10 s stop, and centrifuged at 4 °C, $15,000 \times g$ for 5 mins. The supernatant was reduced and alkylated with 10 mM DTT and 55 mM IAA, respectively. Then the sample was precipitated with chilled acetone (1:4) followed by incubation at $-20$ °C overnight. Then, the precipitant was resuspended in 10 mM Tris HCL, then sonicated for 1 min, with 10-sec pulse and 10 s stop, and centrifuged at 4 °C, $15,000 \times g$ for 5 mins. The supernatant was collected and used for subsequent analysis.

Using CytoBuster$^{TM}$ extraction reagent, a total of 20 female adult *Ae. aegypti* mosquitoes in each replicate were homogenized in 600 µL of the reagent in a mini bead beater using 0.5 mm zirconia beads at 50 rpm for 5 mins at room temperature, and the pellets were then centrifuged at $16,000 \times g$ at 4 °C for 5 mins. The supernatant was transferred into a new tube and concentrated $6\times$ using spin columns with 10,000 molecular weight cut-off (MWCO) (GE Healthcare) at $4,000 \times g$, at 4 °C for 30 mins. After that, the $1\times$ final concentration of protease inhibitor (Sigma, Germany) was added to the protein extract from both methods and kept at $-20$ °C.

## Protein separation

After protein quantification was performed by reducing agent and detergent compatible (RCDC$^{TM}$) Protein Assay (Biorad, USA), a total of 20 µg of protein from each biological replicate and method was heated to 99 °C for 5 mins in SDS loading buffer. The protein samples were loaded onto 10% SDS PAGE gel and run at 200-V for 20 mins until they became stacked on the top of the separating gel. Then, the gel was stained in RAMA stain and incubated for 1 h on a rocker. The staining solution was removed and then washed with distilled water for 3–5 times until bright protein bands were visible.

## In-gel digestion

A protein band comprised of the whole protein mixture from each replicate and method, as mentioned before, was excised from the gel into small pieces and put into a 1.5 ml centrifuge tube. Then 200 µL of a destaining solution made from 80 mg ammonium bicarbonate with 20 mL ACN and 20 mL ultrapure water was added to the gel pieces and incubated at 37 °C for 30 mins with shaking at 300 rpm. The solution mixture was discarded and the process was repeated three times to destain the gels completely. After that, 200 µL fresh reducing buffer (10 mM DTT in 100 mM ammonium bicarbonate) was added to cover the gels and incubated at 60 °C for 30 mins. The gels were allowed to cool, and the buffer was removed. Then, 200 µL fresh alkylating buffer (55 mM IAA in

100 mM AMBIC) was added and incubated in the dark for 60 mins, and the solution was discarded. The gel pieces were washed with a destaining solution and kept at 37 °C for 15 mins, shaking at 300rpm, and then the destaining solution was discarded. After reducing and alkylating, 50μL ACN was added to shrink the gel and incubated for 15 mins at room temperature. The ACN was discarded, and the gel was allowed to air dry for 5–10 mins. Then, 30 μL of 12.5 ng/uL trypsin (Promega, USA) was added to the gel pieces, and the tube was covered with a parafilm and incubated at 37° C overnight shaking at 300 rpm. The digestion mixture was transferred to a clean tube and labelled appropriately.

Further extraction was performed, by adding 10 μl of 1% TFA to the gel pieces, incubated for 5 mins at room temperature and shaking at 300 rpm. The extracted solution was added to the digestion mixture. Then, 50 μL ACN was added to the gel pieces and incubated for 5 mins at room temperature with 300 rpm shaking, and the solution was removed and added to the digestion mixture again. After that, 0.1% TFA was added to the gel pieces, incubated for 5 mins with 300 rpm shaking at room temperature and repeated twice. All the supernatants were combined, and speed vacuumed accordingly.

## LC-ESI-MS/MS analysis

Before the sample loading, samples from three replicates of each method were reconstituted with 30 μL Solvent A and centrifuged at maximum speed for 10 mins. Spatial discrimination of the peptide mixtures was achieved by loading 5 μL of the digested peptides and packed into a large capacity chip, 300A, C18, 160 nL enrichment (Agilent) column and 75 um × 150 mm analytical column (Agilent) with solvent A consisting of water with 0.1% formic acid and Solvent B composed of 90% ACN in water with 0.1% formic acid. The gradient pump eluted with 20–80% ACN for 47 mins and at a flow rate of 4μL/min using Agilent 1200 series capillary pump and 0.5 μL/min Nano pump coupled with Agilent 6550 iFunnel Q-TOF LC/MS/MS. The MS parameters used include positive ion polarity, 1900V capillary voltage with fragmentor voltage of 360V, 325 °C gas temperature and 5.0 L/min drying glass flow. MS acquisitions used were ranged from 110 to 3000 m/z for MS scan and 50–3000 m/z for MS/MS scan. PeaksX was used to examine the MS data against Uniprot Mosquito released 2020_01 from Swissprot and TrEMBL databases with a fixed modification on Carbamidomethylation. A quantitative data normalization automatically performed to correct any experimental bias, and the software calibrated to detect protein threshold to <1% FDR (false discovery rate). From the mass spectrometry data, proteins that showed scores -10lgP $\geq$20 ($-(10\log10(P\text{-value})$) and unique peptides $\geq$1 in all replicates were used for the subsequent analysis.

## Functional annotation analysis

The list of all total proteins from each method obtained from the above analysis was assigned functional categories using the Panther Classification System at http://www.pantherdb.org.

**Table 1  Summary of protein extraction methods.**

| | TCA acetone precipitation method | CytoBuster™ extraction reagent |
|---|---|---|
| Protein amount (µg) mean ± SD | 283.13 µg ± 255.49 | 475.87 µg ± 164.21 |
| No. of proteins identified | 890 | 1,290 |
| Time | 50 hrs | 1 h |

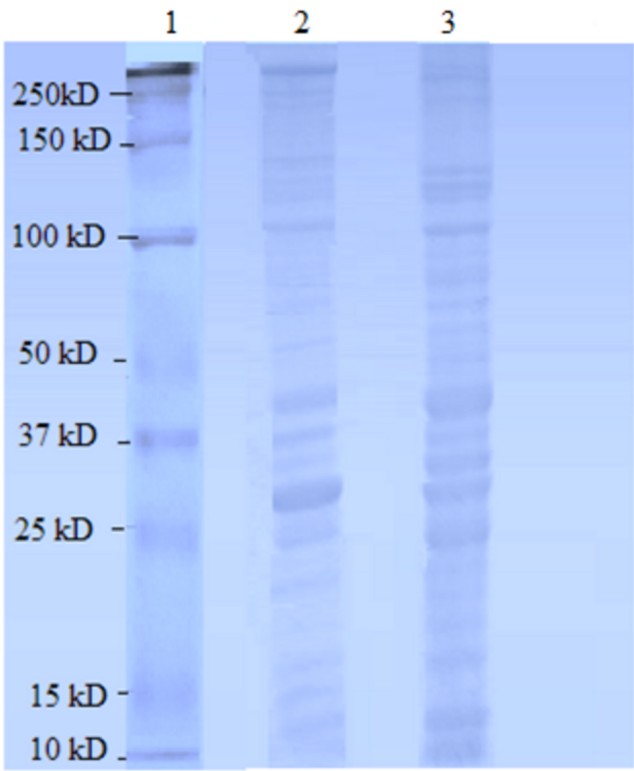

**Figure 1  Female *Ae. aegypti* proteins separated by 10% SDS-PAGE.** *Lane 1* Protein ladder (Precision Plus Protein™ Bio-Rad), *Lane 2* Proteins extracted with TCA acetone precipitation method and *Lane 3* Proteins extracted with CytoBuster™ reagent.

## RESULTS

Analysis of the total protein yield from three biological replicates showed that TCA acetone precipitation extraction and the CytoBuster™ reagent yielded means 283.15 µg and 475.90 µg of proteins, respectively (Table 1).

The protein separation profile by 10% SDS-PAGE revealed the protein bands patterns in female *Ae. aegypti* protein extracted with CytoBuster™ reagent and TCA acetone precipitation method (Fig. 1). There was dissimilarity pattern of protein bands between 250 kDa–100 kDa indicated from both methods. However, more intense protein bands were observed in protein extracted using the CytoBuster™ reagent between 100 kDa–10 kDa though the bands pattern were similar in both methods, as shown in Fig. 1. In the
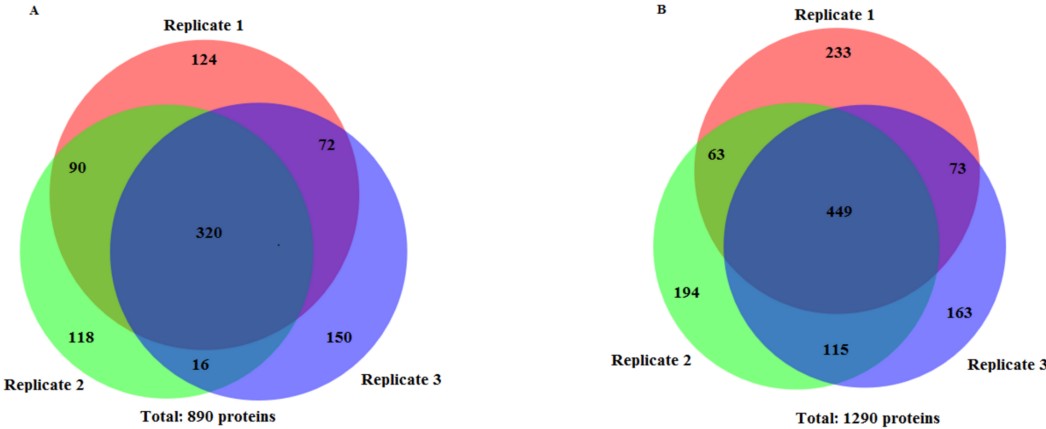

**Figure 2** **Number of identified proteins using two extraction methods.** (A) TCA acetone precipitation extraction method (B) CytoBuster<sup>TM</sup> protein extraction reagent. Ven diagrams were generated using BioVenn (https://www.biovenn.nl/index.php).

TCA acetone precipitation method and the CytoBuster<sup>TM</sup> reagent protein extracts, a total of 890 and 1,290 proteins were identified by LC-ESI-MS/MS, respectively (Fig. 2 A and B). Analysis of LC-ESI-MS/MS of the CytoBuster<sup>TM</sup> reagent extract showed the highest number of identified proteins than the TCA acetone precipitation method by Wang et al. A total of 1,797 proteins were identified by combining both methods, as shown in Fig. 3. Meanwhile, Fig. 4 shows the combination of proteins identified in three replicates from each extraction method. LC-ESI-MS/MS data of this study is available at proteomeXchange with identifier number PXD019698.

There were eight categories of protein class from the proteins identified by LC-ESI-MS/MS extracted using TCA acetone extraction method. The most abundant protein class was the metabolite interconversion-enzyme class (PC00262), with 34.4% hit (11 proteins) (Fig. 5A). The least abundant protein classes were calcium-binding protein class (PC00060), chaperone protein class (PC00072), and nucleic acid-binding protein class (PC00171) with 3.1% hit (1 protein) in each category, respectively (Fig. 5A). There were 11 protein classes from the proteins identified using CytoBuster<sup>TM</sup>. The most abundant protein class was also belonging to the metabolite interconversion-enzyme protein class (PC00262) with 33.3% hit (15 proteins) (Fig. 5B). The least abundant classes were chaperone protein class (PC00072), protein-binding activity modulator protein class (PC00095), nucleic acid-binding protein class (PC00171), a calcium-binding protein class (PC00060), intercellular signal molecule protein class (PC00207) and transporter protein class (PC00227) with 2.2% hit (1 protein) each (Fig. 5B).

## DISCUSSION

This study evaluated two protein extraction methods to analyze female *Ae. aegypti* proteome using LC-ESI-MS/MS. Among the two extraction methods, CytoBuster<sup>TM</sup> extraction reagent was the fastest extraction method with high protein yield (Table 1). In contrast

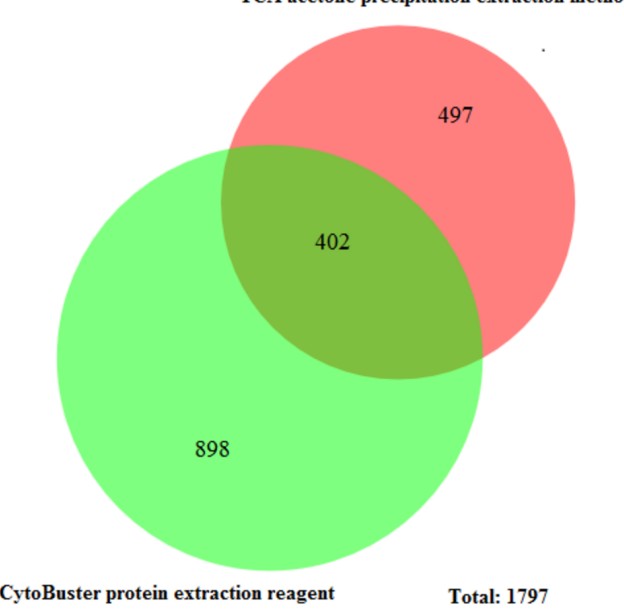

**Figure 3** Unique and common proteins identified from TCA acetone precipitation and CytoBuster™ extraction methods. Ven diagram was generated BioVenn (https://www.biovenn.nl/index.php).

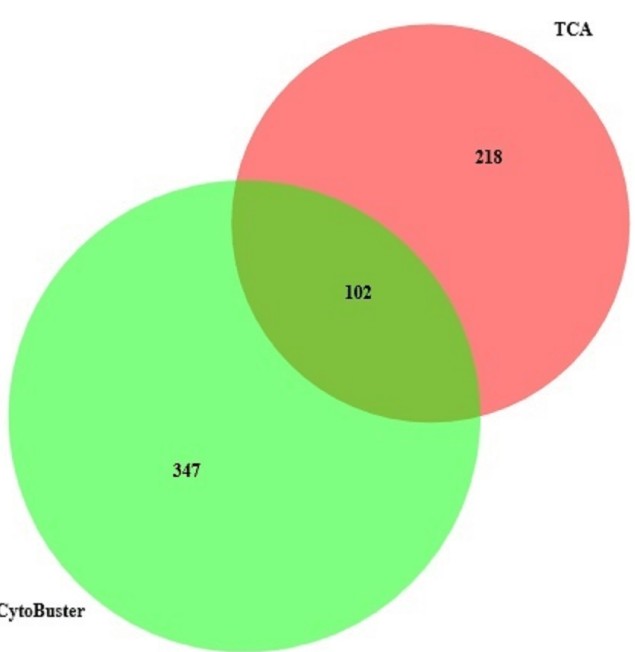

**Figure 4** Protein identified in three replicates using TCA acetone precipitation and CytoBuster™ extraction methods.

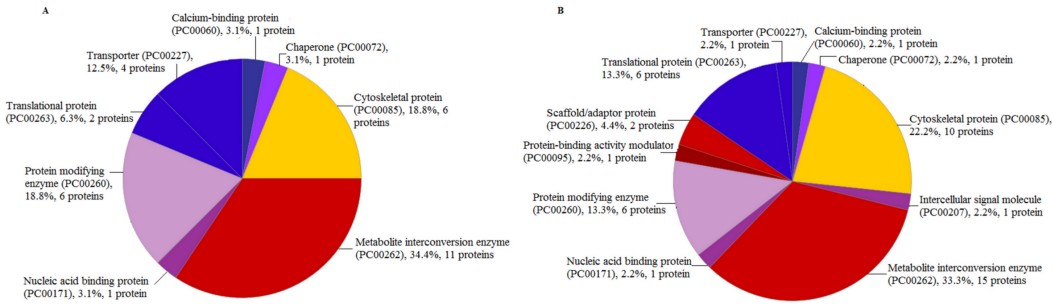

**Figure 5** **Protein Class categories using Panther version 15.0 released 2019-04.** (A) TCA acetone precipitation extraction method. (B) CytoBuster™ protein extraction reagent.

to this evaluation, *Cilia et al. (2009)* reported that TCA acetone precipitation extraction method performed on Aphid produced high protein yield of 20.4 mg/g when compared to phenol and multi-detergent extraction methods yielded 7.3 mg/g and 4.79 mg/g protein, respectively. In addition, protein identification analysis showed 188, 180 and 143 identified proteins from extracts by TCA acetone precipitation, phenol and multi-detergent methods. Similarly, Hassan et al. performed *Plutella xylostella* protein extraction using five different protocols. These included TCA acetone with 0.7% 2-mercaptoethanol, TCA acetone with 2% 2-mercaptoethanol, TCA acetone with 40 mM DTT, lysis buffer (7M urea, 2M thiourea, 4% CHAPS) Tris HCL method and PBS method. They concluded that TCA acetone with 40 mM DTT yielded high total protein amount of 25.17 mg/g with the most abundant protein spots of about 683 (*Hassan et al., 2018*). The TCA/acetone base extraction protocol is one of the most reported protein extraction methods from various samples, including insects. TCA acetone protein extraction method has continued to be an effective method in reducing protein degradation and get rid of interfering elements (*Hassan et al., 2018*).

Visual observation of distinctive protein bands with no smearing showed good quality of protein extract accomplished using both methods in this study (Fig. 1). Hence, the extracts were suitable to be used for the subsequent proteomics analysis. Furthermore, the proteins extracted from CytoBuster™ reagent were recommended, for protein separation using the two-dimensional electrophoresis (2-DE) and Western blotting based on SDS-PAGE analysis where more intense protein bands were obtained.

This study reported the first finding on the comparison of mosquito protein extraction methods between a commercial reagent and TCA acetone precipitation method. The major drawback attributed to the TCA acetone precipitation extraction method was the protein loss due to protein precipitation and resoluble phases required in the process, coupled with many washing steps involved in the technique (*Wu, Gong & Wang, 2014*). At the same time, CytoBuster™ reagent allows the isolation of active functional proteins without the need for additional washing and saves time.

The TCA acetone precipitation method and CytoBuster™ reagent gave 890 and 1,290 proteins identified by LC-ESI-MS/MS, respectively (Figs. 1A and 1B). CytoBuster™ reagent showed the highest number of identified proteins by LC-ESI-MS/MS than the TCA acetone precipitation. The identified proteins from TCA acetone precipitation method

and CytoBuster[TM] reagent represent 5.34%, and 7.75% of the total predicted *Ae. aegypti* proteins, respectively (*Nene et al., 2007*; *Morgat et al., 2020*).

Furthermore, the total proteins identified by LC-ESI-MS/MS covered 10.3% *Ae. aegypti* proteome by the combination of both methods in this study. Overall, we retrieved 29 well-annotated/reviewed, 181 putative, 161 uncharacterized, and 1,398 hypotheticals proteins in this study. Comparatively, *Nunes et al. (2016)* reported a total of 1139 identified proteins from female *Ae. aegypti* heads fed with blood and nectar exclusively, that induced differential protein expression and the identified proteins stood at 7.4% of *Ae. aegypti* proteins. The enriched number of identified proteins by Nunes et al. was due to off gel separation before LC-MS/MS. There were 402 common proteins identified by LC-ESI-MS/MS in both extraction methods (Fig. 3). The CytoBuster[TM] reagent had the highest number of 898 unique identified proteins, suggesting a better coverage than TCA acetone precipitation method. A list of 20 top proteins using LC-ESI-MS/MS can be found in Supplementary Data (Supplementary file 1).

On the other hand, analyses of proteins presence in three replicates from two extraction methods (Fig. 4), suggested by combining both extraction methods could represent a better protein identification and improved proteome coverage. This was shown by the lower number of common identified proteins compared to the unique proteins of the both extraction methods.

There were eleven protein classes from identified proteins extracted by the CytoBuster[TM] reagent compared to eight protein classes by TCA acetone precipitation extraction. We also identified 45 proteins and 11 protein class in CytoBuster[TM] in contrast to 32 proteins and 8 protein class in TCA acetone precipitation method. Proteins associated with scaffold/adaptor protein (PC00226), protein binding activity modulator (PC00095) and intercellular signal molecule (PC00207) were only present in proteins extracted using CytoBuster[TM] reagent (Fig. 5A and 5B).

Among the unique proteins revealed by CytoBuster[TM] reagent is a 14-3-3 protein $\epsilon$ (Q7PX08/Q7PX08_ANOGA) that belongs to Scaffold/adaptor proteins. The 14-2-3 proteins function as adapters, activators and repressors regulating signaling pathways in a range of processes such as cell signaling. In *Drosophila melanogaster*, this protein interacts with many regulators of the actin cytoskeleton (*Ulvila, Vanha-Aho & Rämet, 2011*). 14-3-3 protein $\epsilon$ was crucial for bacterial phagocytosis in *Ae. aegypti* and *Ae. albopictus* (*Trujillo-Ocampo et al., 2017*). Serine proteases inhibitor (serpin) AGAP005246-PA (Q8WSX7/Q8WSX7_ANOGA) was also unique to CytoBuster[TM] reagent extracted proteins. They are acute phase response molecules and regulate immune pathways for human pathogen transmission, and they belong to protein binding activity modulator (*Gulley, Zhang & Michel, 2013*).

## CONCLUSION

We performed a comparative analysis of two different protein extraction methods on female *Ae. aegypti*. CytoBuster[TM] reagent displayed a superior performance for the extraction of proteins in terms of protein yield, the proteome coverage identified by

LC-ESI-MS/MS and extraction speed. The functional analysis also revealed more proteins, and functions attributed to protein extracted using the above reagent based on the rotein class. Nevertheless, combination of two extraction methods could improve the proteome coverage in the presence study.

### Funding
This work was supported by Universiti Sains Malaysia, Research University Individual Grant (RUI) CIPPM.1001.8012228. The funders had no role in study design, data collection and analysis, decision to publish, or preparation of the manuscript.

### Grant Disclosures
The following grant information was disclosed by the authors:
Universiti Sains Malaysia, Research University Individual Grant (RUI): CIPPM.1001.8012228.

### Competing Interests
The authors declare there are no competing interests.

### Author Contributions
- Abubakar Shettima performed the experiments, analyzed the data, prepared figures and/or tables, and approved the final draft.
- Intan H. Ishak and Hadura Abu Hasan conceived and designed the experiments, authored or reviewed drafts of the paper, and approved the final draft.
- Syahirah Hanisah Abdul Rais performed the experiments, prepared figures and/or tables, optimize the experimental protocol, and approved the final draft.
- Nurulhasanah Othman conceived and designed the experiments, analyzed the data, authored or reviewed drafts of the paper, and approved the final draft.

### Data Availability
LC-ESI-MS/MS data of this study is available at the proteomeXchange: PXD019698.
http://proteomecentral.proteomexchange.org/cgi/GetDataset?ID=PXD019698.

### Supplemental Information
Supplemental information for this article can be found online at http://dx.doi.org/10.7717/peerj.10863#supplemental-information.

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
