# Peer review of "Evaluation of female Aedes aegypti proteome via LC-ESI-MS/MS using two protein extraction methods"

_PeerJ, doi:10.7717/peerj.10863_

## Round 0.1 · original submission · Major Revisions

Please carefully address all the critiques of both reviewers and amend your manuscript accordingly.

Reviewer 1 ·

Basic reporting

The authors performed a comparative analysis of two protein extraction procedures: the commercial product Cytobuster and a TCA/acetone extraction, commonly used in the field of proteomics/insect proteomics.
The authors show that more proteins and more protein identifications were obtained in the samples treated and extracted with Cytobuster. Additionally, the extraction protocol with the commercial product was considerably faster than the TCA/acetone protocol used in this study.
The manuscript, however contains a number of problems, such as the selection of extraction methods for the comparison, the confusing description of the instruments used in the LC/MS analysis, lack of information on data analysis, presentation of the results and functional analysis.
Broader and more recent reference set should be used to exemplify past and current developments in the area of proteomics and insect proteomics presented in the manuscript Intro.
Authors should review English used throughout the manuscript.
The authors should include a list with all proteins identified in their runs. Also, this list should be comprehensive, including peptide sequences, number of id’ed peptides, unique peptides, number of spectral counts.
Figures should be revised, the font used for the numbers displayed in the Venn diagrams in figure 1 and the proteins categories in the pie charts of figure 3 is too small.

Experimental design

Material and Methods – Protein Extraction
The two protein extraction methods selected for the comparison presented in this manuscript are completely different.
Though TCA/acetone precipitation is widely used for extraction of proteins, the method as described in the manuscript involves an initial precipitation step with TCA/acetone, a protein pellet resuspended in a lysis buffer without specified composition (line 112), reduction and alkylation and a second precipitation. The two precipitation procedures sum up a 50-hour extraction time described by the authors.
The method used for comparison is commercial alternative optimized for speed, that uses a proprietary solution containing salts and detergents that is going to lyse tissues/cells and extract proteins in a final solution/supernatant. Authors describe a sample concentration step in a spin column “until appropriate concentrations were achieved” (line 125), but do not mention concentrations of what. Also, the concentration of 100x (line 123) of the stating volume of 600 uL would result in 6 uL. Is that correct?
Why did the authors select two extraction procedures so different? Wouldn’t the comparison be fairer and more informative to the field if they also include additional extractions that are detergent-based and without precipitations, similar to what Cytobuster does?
Why was a protease inhibitor cocktail added at the end of both extraction methods instead of at the beginning?
Why was a SDS-PAGE step included at the end of the extractions? What the authors were trying to remove?
Also, note that at this point, TCA/acetone samples were alkylated twice.

Material and Methods – in gel digestion
The description of the final steps of the peptide extraction is very confusing. An “extraction solution” (line 155) used has no description.

Materials and Methods – LC-ESI-MS/MS
Are buffer A (line 164) and solution A (line 166) the same?
In line 165, “… loading … digested peptides and packed into a C18 column”. Could the authors clarify that statement? Is it a pre -column? A pre packed column? A commercial coilumn?
What are the specific dimensions of the C18 column used?
The flow rate described in the method is 4 uL/min but the interface between LC and mass spectrometer is a nano flow Chips/MS interface. Please clarify, either this is not true nanoflow or the flow rate is incorrectly listed in the methods. Also, Agilent 1200 is a microflow LC, how are the authors running a nanoflowchip with this LC system?
The instrument used is an Agilent 6550 iFunnel Q-TOF with a liquid chromatography and ESI source, but the details for mass spectra acquisition are from an LDI instrument. Please clarify, is this LC separation or MALDI?? There are mentions of number of laser shots per spot for MS1 and MS2 which imply this is a MALDI experiment not an LC experiment as previous methods suggest. Please, clarify.
The authors do not provide enough details on the parameters used for data analysis, such as precursor mass tolerance and fragment mass tolerance or any other search parameter. Please add more details.

Validity of the findings

What is the standard deviation of the protein yield for both methods? (table 1)
Authors mention total number of proteins for each method. Could the authors comment on the low reproducibility within methods for the MS runs shown in Fig 1?
The two methods have an overlap of only 402 protein ids. Could the authors provide an explanation for the low overlap? Do the unique proteins for each method share any specific chemical or biological characteristic?
Could the authors provide more information for the functional characterization shown in the manuscript? Only 32 proteins in 8 categories and 45 proteins in 11 categories were assigned for TCA/acetone and CytoBuster, respectively? What about the other proteins? Also, the number of proteins and categories in Result section (lines 197 to 208) and figure 3 do not match the numbers presented in the discussion section (lines 251 to 256).
Discussion
What was the protein yield obtained by the authors? How is compares to the literature mentioned in lines 214 to 216.
The two methods presented showed different yield and different proteins identified, but what do the two extracts look like? Did the author run a SDS-PAGE to evaluate the performance of the two procedures qualitatively?
How do the data presented here compares to other Ae. aegypti whole body proteomes? Is the proteome coverage of 10% good?

Reviewer 2 ·

Basic reporting

In this manuscript, the authors did comparative analysis of two different protein extraction methods in female Aedes aegypti proteome (TCA acetone precipitation extraction method and a commercial protein extraction reagent CytoBuster). It is followed by protein identification using LC-ESI-MS/MS. They identified Cytobuster reagent gave highest protein yield, proteome coverage and extraction speed. They substantiated their findings by performing LC-ESI-MS/MS along with functional annotation analysis in these two protein extraction methods.

Experimental design

No comment

Validity of the findings

1. Protein extraction reagent CytoBuster has also added advantage for performing Westeren blots and SDS page. Do authors observed this advantage while performing their SDS page gel or they don’t see any advantage between these 2 methods for performing SDS page gel in female Aedes aegypti proteome.
2. Do they think Cytobuster has more advantage for performing Western blots or 2D gel electrophoresis? They can add some discussion regarding this. They gave some explanation regarding drawbacks attributed to the TCA acetone precipitation extraction method in their discussion.
3. It would be interesting for the readers to see images of SDS page gel in these 2 methods for comparison. Is there any difference or not? They can show which protein bands or spots are used for in gel digestion.

Additional comments

Overall, in this manuscript the authors supported their findings of CytoBusterTM reagent displayed a superior performance for the extraction of proteins and proteome coverage. The paper is well written and the data supports the observations and conclusions drawn. The work presented in this manuscript, is a useful tool for future studies in the field of proteomic analysis. I hope my comments will be helpful.

---

## Round 0.2 · Minor Revisions

It seems that one extra analysis is still needed: authors find 320 proteins are identified in three replicates of TCA extraction, and 449 are identified in all replicates extracted using CytoBuster. In a followup graph, authors further state that 402 proteins are detected both in TCA extraction and in CytoBuster extraction. This means that at least 449-402= 47 proteins detected in all Cytobuster replicates were not detected in any of the replicates extracted with TCA, which is not surprising due to apparent superiority of Cytobuster. However, authors do not state whether those common "hits" include all (or most) of the 320 proteins that are identified in all replicates using TCA extraction. If a substantial fraction of the proteins extracted in all TCA replicates turn out to not be detected by CytoBuster, this might be a hint suggesting that a protocol combining Cytobuster extraction and TCA extraction could be more effective than any of them in isolation.

Reviewer 2 ·

Basic reporting

The revision has addressed all the concerns well

Experimental design

The revision has addressed all the concerns well

Validity of the findings

The revision has addressed all the concerns well

Additional comments

The authors have done a good job in addressing all the reviewer comments and revising the paper.

---

## Round 0.3 · accepted · Accept

All remaining issues were addressed and I am pleased to accept revised manuscript.